# Transparency as a Means to Analyse the Impact of Inertial Sensors on Users during the Occupational Ergonomic Assessment: A Systematic Review

**DOI:** 10.3390/s24010298

**Published:** 2024-01-04

**Authors:** Marco A. García-Luna, Daniel Ruiz-Fernández, Juan Tortosa-Martínez, Carmen Manchado, Miguel García-Jaén, Juan M. Cortell-Tormo

**Affiliations:** 1Department of General and Specific Didactics, Faculty of Education, University of Alicante, 03690 Alicante, Spain; juan.tortosa@ua.es (J.T.-M.); carmen.manchado@ua.es (C.M.); m.garciajaen@ua.es (M.G.-J.); jm.cortell@ua.es (J.M.C.-T.); 2Department of Computer Science and Technology, University of Alicante, 03690 Alicante, Spain; druiz@ua.es

**Keywords:** transparency, IMU, inertial, accelerometer, ergonomics, RULA, wearable, implementation

## Abstract

The literature has yielded promising data over the past decade regarding the use of inertial sensors for the analysis of occupational ergonomics. However, despite their significant advantages (e.g., portability, lightness, low cost, etc.), their widespread implementation in the actual workplace has not yet been realized, possibly due to their discomfort or potential alteration of the worker’s behaviour. This systematic review has two main objectives: (i) to synthesize and evaluate studies that have employed inertial sensors in ergonomic analysis based on the RULA method; and (ii) to propose an evaluation system for the transparency of this technology to the user as a potential factor that could influence the behaviour and/or movements of the worker. A search was conducted on the Web of Science and Scopus databases. The studies were summarized and categorized based on the type of industry, objective, type and number of sensors used, body parts analysed, combination (or not) with other technologies, real or controlled environment, and transparency. A total of 17 studies were included in this review. The Xsens MVN system was the most widely used in this review, and the majority of studies were classified with a moderate level of transparency. It is noteworthy, however, that there is a limited and worrisome number of studies conducted in uncontrolled real environments.

## 1. Introduction

Excessive load on the musculoskeletal system during certain tasks represents a major occupational ergonomic problem that can trigger various musculoskeletal disorders (MSDs) [1]. MSDs are currently the most prevalent disorders worldwide [2,3]. They form the majority of occupational pathologies, explaining countless causes of work disability, loss of work time, and, consequently, early retirement [4,5]. MSDs can be considered to increase wage compensation and work-related medical expenses, as well as to reduce productivity and quality of life [6,7].

The body areas most frequently affected by these work disorders are generally the lower back, neck, shoulders, elbows, forearms, wrists, and hands [8,9,10]. According to the literature, the best incidence prevention and/or reduction strategy is to minimise exposure to MSD risk factors [11,12]. For this reason, an essential field in occupational ergonomics is the identification of high-risk postures during the workday [13], since correcting and/or adapting these postures diminishes MSD risks and effectively improves work performance [14,15].

Understandably therefore, a large amount of time and effort has been invested in assessing the risk of improper workplace posture across most job sectors [16,17,18]. Many methods exist but Burdorf and van der Beek [19] grouped them into three basic categories: (i) those based on subjective judgments (e.g., questionnaires and measurement scales); (ii) those measured directly; and (iii) systematic observation methods. Discarding the first group owing to their subjectivity, the methods based on direct measurement appear to be the most accurate and reliable [20], although they require significant means to be correctly implemented. Finally, systematic observation methods offer certain advantages such as: simplicity (e.g., a video recording of the worker and, a posteriori, the completion of a series of items based on the recording); greater flexibility (i.e., less—or no—interference with the performed work tasks), and a lower financial cost [21,22]. For these latter reasons, observational methods are the most widely implemented in many work sectors [17,18,23].

Owing mainly to the specificity of the method itself, certain observational methods may be more advantageous or appropriate than others in certain occupations [16]. Yet, we can generally consider that the three most commonly used observational methods are [24] the Ovako Working Posture Analysis System [OWAS] [25], the Rapid Upper Limb Assessment [RULA] [26], and the Rapid Entire Body Assessment [REBA] [27]. Among these three methods, RULA has been defined as perhaps one of the most cautious regarding postural risk assessment [6]. That is, the RULA generally performs a more sensitive posture assessment than the other two methods in the case of most industries, jobs, or postures. In other words, the other two methods tend to underestimate the risks more than RULA [6,16,28,29,30,31].

The use of systematic observation-based occupational risk assessment methods has increased significantly over the past two decades [32], but these methods present limitations that must not be ignored [23,33,34,35]. First, they rely on the visual inspection of certain tasks or procedures by one or more workplace ergonomics professionals, who measure and/or estimate the risk values of each factor under study [23]. This obviously implies a financial expense, so for cost-efficiency reasons, there is always a much smaller number of observers than workers under examination. Moreover, the analysis process is tedious and sometimes imprecise [23]. Thus, occupational risk assessments are usually ultimately limited to the execution of certain repetitive and periodic tasks in relatively controlled and unrealistic environments. Second, in practice, there are variations in the way the different occupational risk tools are implemented [33]. This latter fact is often overlooked, making it difficult to repeat the study or reproduce the results. Third, many work environments are dynamic and continuously change, presenting an ever-increasing heterogeneity of workers, job types, and workplaces [33]. Consequently, occupational risk assessments often need to be conducted more frequently to make them sensitive to these changes. Last but not least, fatigue can cause worker kinematic alteration and/or reduce their control when performing certain tasks [34,35], which is not easily observable with the human eye.

Clearly therefore, to improve occupational health and safety, it is necessary to precisely and objectively quantify the risk factors associated with MSDs [20]. To overcome the systematic observation-based method limitations mentioned above, alternatives based on wearable inertial sensors have recently been advanced. Inertial sensors consist of electromechanical instruments that typically combine an accelerometer, gyroscope, and magnetometer, and are capable of directly measuring linear acceleration and rotational velocity in space. They are arguably an extension of the single-axis inclinometers that have been used by ergonomics and physiotherapy professionals for decades to determine the angulation of any joint structure in the human body. Inertial sensors have been widely used in the scientific literature for a broad range of objectives and applications, from their use as simple, higher-accuracy inclinometers [36], to serving as estimators of physical activity energy expenditure [37]. They have also served to directly evaluate kinematic workplace information, like workers activity [38], physical workload [39], or physical fatigue [40]. The occupational ergonomics sector has highlighted the significant role of this technology in identifying physical risks in the workplace [41,42].

In this latter domain, wearable inertial sensors are today a possible solution with respect to systematic observation-based methodologies in occupational risk assessment [43]. The advantages of the technology have been widely demonstrated, the most notable being its portability, lightness, and low cost [44]. Moreover, the sensors are non-invasive, and the battery lasts for the whole working day, providing more objective, reliable, and consistent results than observational or questionnaire-based methods [45,46]. However, inertial sensors have not yet been adopted on a large scale by occupational risk companies, and recent literature has suggested that this is due to some of their limitations [43,45,47,48]. First, these types of sensors are designed to repeatedly record acceleration or angular velocity in space, though they are unable to interpret the situational context of the recorded tasks [47]. Therefore, to identify the performed task type, additional methods are almost always required, e.g., direct observations, self-measurements, specific additional sensors, etc. [43]. Second, once the task type has been identified, the complexity lies in determining the relevant kinematic and biomechanical characteristics, such as the recorded gesture duration and number of repetitions. In this sense, inertial sensors are presently incapable of determining the start and end of the task on their own. In addition, the repetition count is usually based on postural thresholds linked to joint angulations [43,45], thus also failing to consider relevant factors such as the mobilised load [48]. Third, it is difficult to reproduce the published literature results reliably and objectively owing to the large number of sensors used in some studies (up to 17) [43], as well as the lack of an end-to-end data analysis [43,47].

Nevertheless, despite these limitations, it is hard to explain today why the technology has not become massively used given its advantages. Potential explanations include concerns about its shortcomings and associated discomfort, or because the system could distract or burden workers [41,42,49]. Some of these reasons may be due to suspicions that workers may alter their usual behaviours when they perceive that they are being monitored by inertial sensors. This possible low-level of user transparency has not yet been analysed in the literature, although we believe that it could play a key role in promoting the large-scale implementation of this technology.

Therefore, the aim of this systematic review was twofold: (i) to synthesize and evaluate studies that have used inertial sensors for ergonomic analysis based on the RULA method; and (ii) to advance the “transparency” construct and a way of assessing it based on a range of elements. We also sought to analyse transparency and its possible relationship with other factors in the field of inertial technology as an occupational risk assessment method. We focused only on the analysis of RULA method-based postural evaluations, primarily to avoid biased comparisons between methods that analyse different body parts. Additionally, we considered that the number of available studies that have utilized this evaluation method was sufficient for an initial approach to this new way of analysing system transparency on users.

## 2. Materials and Methods

### 2.1. Search Strategy

The present systematic review was conducted following PRISMA (Preferred Reporting Items for Systematic Reviews and Meta-Analyses) guidelines. Two major social science electronic databases (Web of Science and Scopus) were used to search for relevant publications up to 1 March 2023. Different combinations of keywords and synonyms of the latter were entered in the title, abstract, or keyword search fields: (“inertial” OR “IMU” OR “accelerometer” OR “wearable”) AND (“RULA” OR “Rapid Upper Limb Assessment”). In addition, reference lists of retrieved studies were manually reviewed to identify potentially eligible studies that were not detected in the electronic database searches.

The inclusion criteria were as follows: (i) English as a vehicular language; and (ii) the study had to report the sample size as well as the type, number, and location of all sensors used. We excluded (i) items using deformable or flexible sensors; (ii) proceedings papers; (iii) those with no full-text access; (iv) book chapters; and (v) review articles and/or meta-analyses. The detailed screening of the title and abstract of each pre-selected study was conducted independently by two authors (MAG-L and JMC-T). The full texts were reviewed to identify the articles that met the selection criteria. When discrepancies between the two authors appeared in the selection process, a consensus was reached with a third author (DR-F). 

### 2.2. Risk of Bias Assessment

The methodological evaluation process was conducted by two authors (MG-J and CM) using an adapted version of the STROBE assessment criteria for cross-sectional studies [50]. Each article was evaluated based on 10 specific criteria (Table 1) and any disagreement was resolved via consensus. Each of the 10 criteria (or items) was assessed as “1” (fulfilled) or “0” (not fulfilled), and following the criterion of O’Reilly et al. [51], each study was qualitatively categorised as follows: a study was considered of high quality (i.e., low bias risk) when it scored “1” in eight or more criteria, and of low quality (i.e., high bias risk) when it scored “0” in three or more criteria.

### 2.3. Data Extraction

The data was extracted to Microsoft Excel (Microsoft Corporation, Redmond, WA, USA) following the Cochrane Consumers and Communication Review Group’s data extraction template (Cochrane Consumers and Communication, 2016). The Excel spreadsheet was used to evaluate the inclusion criteria and to then verify all selected articles. The process was conducted independently by two authors (MAG-L and JT-M), and any eligibility disagreement was resolved via a discussion with a third author (DR-F). Full-text articles that were excluded were recorded with the reasons for exclusion. All results were stored in the Excel sheet. The following information was extracted from the original articles: (i) industrial sector; (ii) main objective; (iii) type and number of sensors used; (iv) body elements analysed; (v) combination (or not) with other technologies; (vi) real or controlled environment; and (vii) transparency level.

### 2.4. Transparency Assessment

Next, a proposal was advanced to qualitatively evaluate how the different data capture devices affecting users. This impact could positively or negatively modify the development of the test: it could have a positive effect because users could, for example, exert more effort than usual in performing the exercises; and it could have a negative effect if the different devices hindered users in their tasks or conditioned them psychologically.

We consider it relevant to emphasize the specific context in which we are seeking to define the concept of “transparency” to avoid confusion. Transparency can be defined with different and varied meanings, ranging from the absence of deceit to the quality of being easily detectable or visible. Furthermore, given that privacy and confidentiality are relevant issues in assessments based on IMUs [41,52], it could be easy to confuse and assume that the concept of “transparency” presented in this work is related to them—but it certainly is not.

Therefore, we shall use the term “transparency” hereinafter to relate the user task to be performed with the possible device impact on movements or behaviours. Thus, the transparency of a collection system can be defined as the greater or lesser degree to which users are aware of the sensing of their movements. The degree of transparency is therefore inversely proportional to the degree of impact of the data collection system and mainly depends on the following variables:Number of devices. This variable refers to the number of devices included in the data capture system. The greater the number of system devices, the greater the users’ perception that they are being monitored and, therefore, the greater the possible impact. To quantify the variable, a low level was defined as 1 or 2 devices, an average level as 3 to 5 devices, and a high level above 5.Device visibility. The more visible the devices are, the more aware the user will be of them and, therefore, the greater their possible impact. The distribution of this variable was determined as follows: a low level if the devices are not visible; a medium level if the devices are visible, but leave the user freedom of displacement (e.g., they are not restricted to a laboratory); and a high level if the devices are not only visible but also limit user mobility across various situational contexts, rooms, or locations.Device contact. Some sensor elements may not be visible to users, but they are in contact with them. The greater the degree of contact (whether according to surface area or device size, or due to invasive devices), the greater the user’s perception and the greater the risk of being affected by the devices. In this case, the following variable gradation was defined: a low level in the case of no device contact; a medium level in the case of contact with a reduced surface area (e.g., electrocardiography sensors); and a high level in the case of contact with a large surface area or invasive sensors.

These variables can coexist in a capture system and interrelate with each other. To assess the overall system transparency level, we defined one variable as the main one, while the other two acted as transparency level modifiers, either raising or reducing it. The variable we defined as the main variable was visibility. A high impact level of one of the other variables increased the impact of this variable, and a low impact level reduced it. An average level was considered neutral. As an illustration, we can picture a wireless system made up of 3 electrocardiography sensors that connect to a mobile phone via Bluetooth. In this case, the impact of the visibility variable would be low (the sensors located under the clothing are not visible and neither is the mobile phone). Moreover, the contact is also low-impact, further reducing the impact of the system as a whole. Finally, the number of devices is 4 (i.e., 3 sensors plus the mobile phone), so the impact level would be average and would not modify the system’s impact. In this case, the overall transparency would be high (since the impact is low and transparency is inversely proportional to the impact).

**Table 1 sensors-24-00298-t001:** Risk of bias assessment of the included studies based on the modified STROBE criteria.

Study	1	2	3	4	5	6	7	8	9	10	Quality
Battini et al. [53]	1	0	1	1	1	1	0	0	0	1	6
Blume et al. [54]	1	0	1	1	1	1	0	1	1	1	8
Carbonaro et al. [55]	1	0	0	1	1	0	0	1	1	1	6
Colim et al. [56]	1	0	0	1	1	1	1	1	1	1	8
Maurer-Grubinger et al. [57]	1	0	1	1	1	0	1	1	1	1	8
Hokenstad et al. [58]	1	0	0	1	1	1	1	1	1	1	8
Holzgreve et al. [59]	1	0	1	1	1	1	0	1	1	1	8
Holzgreve et al. [60]	1	0	1	1	1	1	0	0	0	1	6
Huang et al. [61]	1	0	0	1	1	1	0	1	1	1	7
Humadi et al. [62]	1	1	0	1	1	1	0	1	1	1	8
Humadi et al. [63]	1	1	0	1	1	1	0	1	1	1	8
Ohlendorf et al. [64]	1	0	1	1	1	1	0	0	0	1	6
Reddy et al. [65]	1	0	0	1	1	0	0	1	1	0	5
Ryu et al. [66]	1	0	1	1	1	1	0	1	0	0	6
Vignais et al. [67]	1	0	0	1	1	1	1	1	1	1	8
Vignais et al. [68]	1	0	0	1	1	1	1	1	1	1	8
Weitbrecht et al. [69]	1	0	1	1	1	1	1	1	0	1	8

Item legend: 1. Present in the abstract a comprehensive and impartial overview of the conducted activities and findings. 2. Express specific objectives, including any predetermined hypotheses. 3. Outline eligibility criteria, along with the sources and methods used for participant selection. 4. For each variable of interest, provide data sources and details regarding assessment methods (measurement). Discuss the comparability of assessment methods if multiple groups are involved. 5. Clarify the approach taken to handle quantitative variables in the analyses. If relevant, elaborate on the chosen groupings and their rationale. 6. Detail the characteristics of study participants (e.g., demographic, clinical, social) and provide information on exposures and potential confounding factors. 7. Summarize key results in relation to study objectives. 8. Delve into the study’s limitations, addressing potential sources of bias or imprecision. Consider both the direction and magnitude of any potential bias. 9. Offer a cautious overall interpretation of results, taking into account objectives, limitations, the multitude of analyses, findings from similar studies, and other pertinent evidence. 10. Disclose the funding source and elucidate the role of funders in the current study and, if applicable, in the original study upon which the current article is based. STROBE: STrengthening the Reporting of OBservational studies in Epidemiology.

## 3. Results

### 3.1. Search Results

The database search produced 83 results (Figure 1). After removing duplicates (n = 36), the title, keywords, and abstract of 47 articles were analysed, and 17 were excluded because they did not meet the inclusion criteria. The eligibility of the rest of the articles (n = 30) was evaluated, and 17 articles were chosen for the final analyses. Among them, 14 were published between 2021 and 2023, while only 3 of them were published between 2013 and 2020, thus reflecting the scientific community’s growing interest in the field.

### 3.2. Risk of Bias Assessment

Table 1 shows the overall assessment of the methodological quality of the included studies. A total of 10 (59%) of the 17 included articles presented high methodological quality, while the remaining 7 (41%) were of low methodological quality. Of the latter, none scored below 5 points in the checklist used. All analysed studies described the data source and detailed the evaluation methods (item 4). In the same way, all included articles explained the management of quantitative variables in the analysis process (item 5). Most studies sufficiently detailed the research participants as well as the study limitations (items 6 and 8, 82%). Likewise, most authors interpreted the results with caution (item 9, 71%) and reported the source of funding (item 10, 88%). Many authors inadequately reported, specified, or justified the objectives with their consequent hypotheses (item 2, 88%). Finally, most articles insufficiently detailed the sample selection criteria (item 3, 53%) and lacked a summary of the key results based on the established objectives (item 7, 65%). The assessments of the two authors were consistent and comparable, with mean review scores of 7.12 ± 0.86 and 7.24 ± 1.03.

### 3.3. Industry Type and Objective

Table 2 summarises the main characteristics of each study included in this review. The analysed works focused on several industries. Among the 17 studies, the most represented sector (n = 6; 35%) was industrial manufacturing and manual material handling tasks [53,56,62,63,67,68]. Four (n = 4; 24%) were dedicated to the dental sector [54,57,60,64], and another four (n = 4; 24%) to surgery [55,58,65,69]. Finally, two studies (n = 2; 12%) were related to the construction sector [61,66] and one (n = 1; 6%) to a more generic field linked to administrative tasks performed on a computer at a desk [59].

Regarding the general objective of the included articles, six (n = 6; 35%) addressed the development and presentation of systems based on inertial systems to analyse one or more ergonomic indices [53,55,57,61,66,67]. Another six studies (n = 6; 35%) designed their intervention based on an ergonomic risk comparison between different workstations or workstation implementations [56,58,59,60,64,65]. Finally, three works were dedicated to analysing ergonomic risk in a specific industry or task [54,68,69], and two others analysed the accuracy and reliability of this technology for workplace ergonomic assessment [62,63].

### 3.4. Technology Used and Body Elements Analysed

Firstly, in relation to the sensors used in each analysed study, the vast majority (n = 13; 77%) used the Xsens MVN system. Of these, most (n = 11) set up all sensors (i.e., 17 MTw sensors) including the MVN system [53,54,57,59,60,61,62,63,64,66,69], and only two used 11 [56] and 3 [55] MTw sensors, respectively, from the Xsens MVN system. On the other hand, in only four cases [58,65,67,68] was the Xsens MVN system not used, a different inertial sensor system being employed in its place. One study was based on the I2M, SXT version, ADPM [58] with four sensors; another on the Opal V2, Mobility Laboratory, ADPM [65] with three sensors; another on the Trivisio GmbH’s Hummingbird IMUs [67], with seven sensors; and, finally, the CAPTIV Motion IMUs from Tea, Nancy, with seven sensors was applied in the last [68].

Secondly, only two studies (n = 2; 12%) combined inertial technology with 3D motion capture instruments (VICON) in order to validate the results obtained with inertial sensors [62,63]. On the other hand, seven studies (n = 7; 41%) used video recording as a reference when contrasting the data [53,54,55,61,66,68,69]. Finally, two studies (n = 2; 12%) required the use of goniometers (bi-axial SG65, Biometrics Ltd., Newport) on the back of the hand due to sensor system limitations [67,68], and seven studies (n = 7; 41%) did not implement any combination with inertial technology [56,57,58,59,60,64,65].

Thirdly, the vast majority of the studies (n = 14; 82%) used protocols focused on analysing the upper body [54,55,56,57,58,59,60,62,63,64,65,67,68,69]. Only three studies (n = 3; 18%) analysed the entire body [53,61,66].

### 3.5. Environment and Transparency for the User

The vast majority of studies (n = 14; 82%) were conducted in a controlled environment, with the intention of simulating a real situation to a greater or lesser extent [53,54,56,57,59,60,61,62,63,64,66,67,68,69]. Only three studies (n = 3; 18%) extracted their results from a real environment in their respective industrial sectors [55,58,65].

Finally, based on the scale proposed in this study, transparency was average across most (n = 12; 71%) studies [54,56,57,58,59,60,61,62,63,64,66,69]. Three papers (n = 3; 18%) presented low transparency [53,67,68], and only two (n = 2; 12%) showed high user transparency [55,65].

## 4. Discussion

The systematic review presented here confirms the relevance of using inertial technology to assess workplace posture. The large number of studies published over the last three years compared to previous years (i.e., 14 vs. 3 studies in the last three years) demonstrates the rising interest in the technology. In this sense, the use of inertial technology to assess posture and its risks in the workplace has demonstrated its numerous advantages such as portability, lightness, and low cost—both on a human and technological level [43,44,45,46]. However, despite its advantages compared to traditional observational methods, its application has not yet become widespread in the workplace. Some of the reasons recently put forward in the literature include the uncertainties surrounding possible discomfort, or low levels of transparency for users [41,42,49,52]. Therefore, the aim of this systematic review was to propose a novel way of assessing the user “transparency” of inertial technology and its possible relationship with other factors in the field of occupational risk assessment.

In this work, we presented a pioneering method to qualitatively categorise inertial capture system transparency to address the hypothesis that it may be affecting or conditioning worker movements or behaviours. In short, transparency would be defined as the degree to which users are aware that their movements are being monitored. The degree of transparency is inversely proportional to the degree of user impact (i.e., low transparency implies that it highly affects the system users, and vice versa). The aim was to relate transparency to the rest of the factors analysed in order to better understand this concept.

The diversity of work sectors found in this review was similar to that reported in other recent reviews [43,45]. All studies conducted in the dental sector [54,57,60,64], the construction sector [61,66], and the computer-based administrative sector [59] reported average user transparency. This average transparency level was also found in half the works conducted in the industrial manufacturing and manual material handling tasks sector [56,62,63], as well as in half of those in the domain of surgery [58,69]. The other half of the studies in the field of industrial manufacturing and manual material handling tasks [53,67,68] and in surgery [55,65] found low and high transparency, respectively. Instruments with average transparency seem to have been mostly used in various professional fields. In this sense, given that the number of used sensors largely depends on the sector and task type under study [43], the form of measurement should be standardised according to task type to obtain more consistent and comparable results. However, it seems unlikely that the type of industry directly determines the transparency of the device to be used, since a range of sectors apply instruments that are varyingly transparent.

Regarding the type of objective of the included studies, the two studies [62,63] that focused on analysing the precision and reliability of the instrument obtained average transparency. Of the three which analysed ergonomic risk in a particular sector or task, one [68] obtained low transparency and the other two [54,69] average transparency. The studies that compared ergonomic risk designs between different workstations [56,58,59,60,64] found that a majority of protocols presented average transparency compared to those with high transparency [65]. Finally, among the six works that developed and presented new inertial systems, low- [53,67], average- [57,61,66], and high-level transparencies [55] were encountered. Thus, the type of objective does not seem to be determining the used system transparency level either. The kinds of objectives observed were similar to that in other recent reviews, and no clear associations with the technology type used appeared [43,45].

As in previous reviews [70,71], the Xsens MVN system seems to be one of the most widely employed, whether all or part of the sensor system is applied. On the one hand, of the 13 studies that used the Xsens MVN system, most implemented 17 MTw sensors, leading to average transparency [54,56,57,59,60,61,62,63,64,66,69], and only 2 obtained low [53] and high transparency [55], respectively. Another work for which average transparency was encountered was the one that used the I2M system, SXT version, ADPM [58]. On the other hand, the studies based on Colibrio IMUs of Trivisio GmbH, or the CAPTIV Motion IMUs of Tea, Nancy, ultimately demonstrated low transparency [67,68], and the study that used the Opal V2 system, Mobility Laboratory, ADPM, achieved high transparency [65]. The type of measuring system or instrument does seem to determine user transparency, since each system is composed of a determined number of sensors and a specific sensing method. In this case, the only two systems that showed high transparency were the Xsens’ MVN and Opal V2 instruments, and both were used in a protocol with three-sensor instrumentation [55,65]. Regarding device transparency categorisation, both protocols shared low visibility and an average amount of sensors and contact. Therefore, though we considered visibility as the main transparency categorisation variable, it seems that the number of sensors could be key to achieving high user transparency. This is especially true given that today at least, it would not be possible to eliminate contact using this type of inertial technology [43,45].

Focusing on transparency in relation to the environment used, of the 14 studies that were conducted in a controlled environment, the vast majority showed average transparency [54,56,57,59,60,61,62,63,64,66,69]. Only a minority presented low transparency [53,67,68]. On the other hand, of the three studies conducted in a real environment, two [55,65] presented a high transparency level and the third [58] an average level. It is rather odd, to say the least, that the only two instruments to have obtained high user transparency were those that were used in real-world protocols. The reduced number of studies in real environments does not allow us to explore any associations, but this subject could be addressed in future studies. We thus suggest the effectiveness of this technology continue to be analysed in real uncontrolled environments, since a lack of studies in this field could be limiting its large-scale implementation.

Considering all of the above, it seems obvious that to increase user transparency, not only should the inertial sensor visibility be reduced as much as possible, but also the number of sensors used [43]. However, this could lead to lowering the quantity and/or quality of the information recorded by the technology. It would thus be necessary to weigh the benefits of increasing user transparency (i.e., to affect their behaviour as little as possible) against the cost of receiving limited and/or reduced information [43]. On the other hand, some of the limitations of this technology could perhaps be solved thanks to promising advances in artificial intelligence, machine learning, and/or neural networks [72]. Finally, regardless of the associations exposed here between transparency and other factors, we cannot conclude whether transparency directly affects worker behaviour or the large-scale applicability of this technology. We suggest that future studies explore whether technology transparency affects worker behaviour, movements, and posture. Moreover, they should evaluate whether this or some other type of user transparency is preventing inertial technology from being implemented in the domain of workplace ergonomic assessment generally.

## 5. Conclusions

In this work, we put forward a qualitative method to assess transparency levels for users in the field of inertial technology-based workplace ergonomic assessment. In addition, we reviewed transparency levels obtained in workplace ergonomics studies that used inertial sensors and the RULA method. The most important ideas that can be extracted from this review are the following:Most of the reviewed studies presented average transparency;The Xsens MVN system was the most widely used in the articles included in this review;Last but not least, a concerning and insufficient number of studies were conducted in real, uncontrolled environments. This shortcoming may be restricting the advancement of knowledge and the large-scale application of inertial technology-based workplace ergonomic risk assessment.

## Figures and Tables

**Figure 1 sensors-24-00298-f001:**
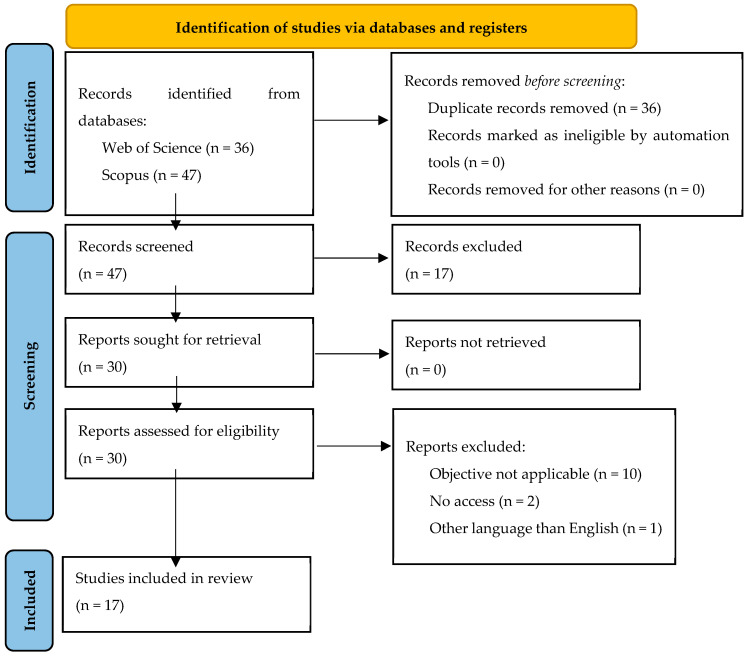
Search strategy Preferred Reported Items for Systematic Reviews and Meta-Analyses (PRISMA) flow chart.

**Table 2 sensors-24-00298-t002:** Characteristics of each study.

Study	Industrial Sector	Main Objective	Type and Number of Sensors Used	Body Elements Analysed	Combination with Other Technologies	Real or Controlled Environment	Transparency
Battini et al. [53]	Furniture manufacturing	To present a platform that evaluates 4 ergonomic indices in real time	Xsens MVN (17 MTw Awinda IMU sensors); G4 MOCAPSUIT (29 Synertial IMU sensors); AIDlab activity tracker (5 sensors: ECG, HR, ST, RP, and MP)	Full body	Validation with video recording (contrasting 10 frames representing the postures)	Controlled	Low
Blume et al. [54]	Dentistry	To analyse the ergonomic risk of dental students	Xsens MVN (17 MTw Awinda IMU sensors)	Upper body	Video as reference	Controlled	Average
Carbonaro et al. [55]	Surgery (laparoscopy)	To present and describe an instrument designed to monitor posture during surgery	Xsens MVN (3 MTw Awinda IMU sensors)	Upper body	Video as reference	Real	High
Colim et al. [56]	Furniture manufacturing	To analyse MSD risk before and after robotic implementation	Xsens MVN (11 MTw Awinda IMU sensors)	Upper body	No	Controlled	Average
Maurer-Grubinger et al. [57]	Dentistry	Methodological development of the quantification of workplace ergonomics based on RULA	Xsens MVN (17 MTw Awinda IMU sensors)	Upper body	No	Controlled	Average
Hokenstad et al. [58]	Surgery (hysterectomy)	To monitor the operation and compare it with another, guided by ergonomic recommendations	Opal, APDM (4 I2M SXT IMU sensors)	Upper body	No	Real	Average
Holzgreve et al. [59]	General, undergraduate students	To compare the ergonomic risk of working at home vs. optimised workplace	Xsens MVN (17 MTw Awinda IMU sensors)	Upper body	No	Controlled	Average
Holzgreve et al. [60]	Dentistry	To investigate ergonomic risk in four dental fields and compare dentists and assistants	Xsens MVN (17 MTw Awinda IMU sensors)	Upper body	No	Controlled	Average
Huang et al. [61]	Construction: shipbuilding	To develop and validate an inertial sensor-based system to assess WMSD risk	Xsens MVN (17 MTw Awinda IMU sensors)	Full body	Video as reference and agreement with experts	Controlled	Average
Humadi et al. [62]	Manual material handling tasks	To investigate the accuracy and reliability of wearable technology and markers (with Kinnect V2) vs. MOCAP	Xsens MVN (17 MTw Awinda IMU sensors)	Upper body	Validation against MOCAP 3D (VICON with 8 cameras and 100 Hz sample frequency)	Controlled	Average
Humadi et al. [63]	Manual material handling tasks	To investigate the accuracy and repeatability of an IMU system for RULA evaluation	Xsens MVN (17 MTw Awinda IMU sensors)	Upper body	Validation against 3D and 2D MOCAP (VICON, 100 Hz)	Controlled	Average
Ohlendorf et al. [64]	Dentistry	To investigate ergonomic risk in four workplace concepts and compare dentists and assistants	Xsens MVN (17 MTw Awinda IMU sensors)	Upper body	No	Controlled	Average
Reddy et al. [65]	Surgery (microsurgery for male fertility)	To compare ergonomic risk using 4K-3D Exoscope vs. traditional operating microscope	Opal V2, ADPM (3 I2M SXT IMU sensors)	Upper body	No	Real	High
Ryu et al. [66]	Construction: bricklaying	To investigate the applicability of RULA, REBA, and OWAS to masonry	Xsens MVN (17 MTw Awinda IMU sensors) and Perception (Notim) (17 Neuron IMU sensors)	Full body	Video as reference	Controlled	Average
Vignais et al. [67]	Industrial manufacturing	To present an innovative and practical ergonomic evaluation system	Trivisio GmbH (7 Colibri IMU sensors)	Upper body	Goniometers on the back of the hand (Bi-Axial SG65 Goniometers, Biometrics Ltd., Newport, UK)	Controlled	Low
Vignais et al. [68]	Manual material handling tasks	To analyse the ergonomic risk of a manual task in combination with video recording	CAPTIV (7 TEA T-Sens Motion IMU sensors)	Upper body	Goniometers on the back of the hand (bi-axial SG65 goniometers, Biometrics Ltd., Newport). Video as reference	Controlled	Low
Weitbrecht et al. [69]	Surgery (oral and maxillofacial)	To analyse the ergonomic risk of this occupational group	Xsens MVN (17 MTw Awinda IMU sensors)	Upper body	Video as reference	Controlled	Average

MSD: musculoskeletal disease; WMSD: work-related musculoskeletal disease; MOCAP: motion capture; IMU: inertial measurement unit; RULA: Rapid Upper Limb Assessment; REBA: Rapid Entire Body Assessment; OWAS: Ovako Working posture Assessment; ECG: electrocardiogram; HR: heart rate; ST: skin temperature; RP: respiration; MP: microphone; IMU: inertial measurement unit.

## Data Availability

Not applicable.

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
