# Peer review of "Transparency as a Means to Analyse the Impact of Inertial Sensors on Users during the Occupational Ergonomic Assessment: A Systematic Review"

_sensors, 2024, doi:10.3390/s24010298_

Round 1

Reviewer 1 Report

Comments and Suggestions for Authors

Transparency as a means to analyse the impact of inertial sen- 2 sors on users during the occupational ergonomic assessment: a 3 systematic review.

The article is suggested to be revised and published.

1.The present situation at home and abroad is not comprehensive enough. Suggested quotation Chang L, Guo Y, Huang X, et al. Experimental study on the protective performance of bulletproof plate and padding materials under ballistic impact[J]. Materials & Design, 2021, 207: 109841.

2.The abstract suggests further refinement.

3.Important conclusions are suggested to be described point by point.

Comments on the Quality of English Language

Extensive editing of English language required.

Author Response

Dear reviewer, thank you very much for your comments and suggestions; they have helped us improve the quality of the work. Below, I will address each of the questions you raised one by one:

1.The present situation at home and abroad is not comprehensive enough. Suggested quotation Chang L, Guo Y, Huang X, et al. Experimental study on the protective performance of bulletproof plate and padding materials under ballistic impact[J]. Materials & Design, 2021, 207: 109841.

Answer: After carefully reviewing the experimental study that you suggested on the protective performance of bulletproof plate and padding materials under ballistic impact, we have not found a way to include it in our work to better describe the situation, because we have been unable to find the connection between the two works. Nevertheless, we have added a new reference in an attempt to provide further insights into the data privacy context and clarify the definition of the term "transparency" proposed in our manuscript (lines 186-192).

2.The abstract suggests further refinement.

Answer: The abstract has been revised and refined (lines 13-27).

3.Important conclusions are suggested to be described point by point.

Answer: We have rewritten and described the conclusions point by point as you suggest to make them more structured and easy to read (lines 464-472).

NOTE: Regarding the English translation, we have translated the work through a specialized service with professional native English-speaking translators. We are attaching the certificate in PDF.

Reviewer 2 Report

Comments and Suggestions for Authors

The review content is not good. The introduction does not get into the main topic after about 2/3 page space. The current reviewer did not get any conclusive knowledge from the submission.

Technically, is it appropriate to put the method of using inertial sensors as the category of observations (instead of directly measuring)?

Comments on the Quality of English Language

Fair.

Author Response

Dear reviewer, thank you very much for your comments and suggestions; they have helped us improve the quality of the work. Below, I will address each of the questions you raised one by one:

1.The review content is not good. The introduction does not get into the main topic after about 2/3 page space. The current reviewer did not get any conclusive knowledge from the submission.

Answer: After reviewing the introduction, we believe that providing a detailed contextualization of the various topics leading to the problem statement and research objective is important. When describing the lack of standardization of inertial sensors in the field of occupational ergonomics, we believe that before delving into the technological section, it is important to mention the relationship between occupational ergonomics and musculoskeletal disorders. Additionally, we also consider relevant to discuss the various traditional forms of evaluation that are commonly used (along with their main disadvantages compared to new tools). However, if you feel that eliminating/rewriting any section would enhance the introduction, we would be happy to do so. Additionally, we have rewritten the conclusions, presenting them point by point to make them more straightforward for the reader to understand (lines 464-472).

2.Technically, is it appropriate to put the method of using inertial sensors as the category of observations (instead of directly measuring)?

Answer: It's possible that we did not express ourselves correctly in the paper, but at no point did we attempt to include inertial technology as a category of observations. We have treated it as an alternative to more commonly used methods, which are based on systematic observation. We have mentioned it as a solution to the problems that this type of methodology presents when used, not as an additional category within systematic observation. As you rightly point out, if we were to include sensors in any category in the assessment of occupational risks, it would be within the one that involves direct measurement. Although, since inertial sensors are being used as a facilitating means in the administration of observational methods (e.g., RULA, REBA, etc.), we do not specifically classify them within any category in our text.

NOTE: Regarding the English translation, we have translated the work through a specialized service with professional native English-speaking translators. We are attaching the certificate in PDF.

Reviewer 3 Report

Comments and Suggestions for Authors

I suggest that this paper needs major revisions, particularly in the clarity of certain terms, aims, and the rationale behind selecting certain assessment methods. Given the shortcomings of this paper, its significance and importance to the field are questionable. These are some examples of the issues that I found with the paper that I believe need to be resolved:

In the introduction section, it is not clear why RULA was used in the analysis. While it is claimed to be more accurate than other observation methods, it is unclear why it should be a criterion for selecting IMU-based studies.

Additionally, the term "transparency" can be confusing. Transparency can indicate both being free of deceits, while also referring to something that is easily detectable or seen, Especially in the context of this paper, as privacy and confidentiality concerns exist when using IMU-based assessments ( [42] on the reference list), an alternative term or clearer definition of “Transparency” in the Introduction section would be useful.

More information is needed on how the criteria and scales were decided for transparency assessment. What were the rationales? They appear to be arbitrary. Also, providing examples would be helpful. For instance, what is an example of a low-level no-contact IMU?

Line 356: IMU can be considered a low-cost motion-capturing system compared to others, but it is not as low-cost as observational studies such as RULA.

In lines 359 and 427-428 you would see the issue with using the term “transparency” again. Based on line 359, the problem with IMU can be “lack of transparency”? but in lines 427-428 it has been mentioned that to “improve transparency” (which can infer as to increase transparency) fewer sensors should be used, which may confuse the readers.

Comments on the Quality of English Language

Good.

Author Response

Dear reviewer, thank you very much for your comments and suggestions; they have helped us improve the quality of the work. Below, I will address each of the questions you raised one by one:

1.In the introduction section, it is not clear why RULA was used in the analysis. While it is claimed to be more accurate than other observation methods, it is unclear why it should be a criterion for selecting IMU-based studies.

Answer: Dear reviewer, we fully agree that the ergonomic evaluation methods (e.g., RULA, REBA, OWAS, etc.) should not be used as criteria for selecting IMU-based studies, and we have not done so either. Since our study focuses on proposing a new scale to assess transparency on users, we deemed it more appropriate to narrow the search to a single method to avoid biased comparisons. Considering that RULA primarily focuses on the analysis of the trunk and upper limbs, it would not be fair to include transparency comparison methods that involve more body parts, and probably requiring a higher number of sensors. RULA is one of the most widely used and accurate methods to date. Given its extensive use in numerous studies, we believe it provides a sufficient basis for introducing our new proposal. In any case, we have briefly outlined these reasons at the end of the introduction to inform the reader and make our choice more understandable (lines 135-139).

2.Additionally, the term "transparency" can be confusing. Transparency can indicate both being free of deceits, while also referring to something that is easily detectable or seen, Especially in the context of this paper, as privacy and confidentiality concerns exist when using IMU-based assessments ( [42] on the reference list), an alternative term or clearer definition of “Transparency” in the Introduction section would be useful.

Answer: We have included the information you recommend in the definition of the term "transparency" to assist the reader and prevent any potential confusion (lines 186-192).

3.More information is needed on how the criteria and scales were decided for transparency assessment. What were the rationales? They appear to be arbitrary. Also, providing examples would be helpful. For instance, what is an example of a low-level no-contact IMU?

Answer: The criteria we have established to determine the scale and importance of the variables composing it stem from the concept we have defined as "transparency." Based on these criteria and variables, we have structured and prioritized them according to their significance in the definition. We acknowledge that there may always be a certain degree of arbitrariness in creating new terms or constructs. Nevertheless, we also believe that these new contributions help achieve a more holistic and integrated understanding of a specific topic in relation to its associated factors. In relation to the examples you suggest, we consider them important, and in fact, we have included some in our text (lines 223-231). Furthermore, regarding the specific example you propose (low-level no-contact IMU), it is something for which technological capability is not currently available. This, precisely, is one of the ideas mentioned in the discussion (lines 428-433).

4.Line 356: IMU can be considered a low-cost motion-capturing system compared to others, but it is not as low-cost as observational studies such as RULA.

Answer: To some extent, the authors agree with your statement. However, we would like to clarify that when we talk about "low-cost," we are not only referring to the cost of purchasing or maintaining the instrument. Additionally, we must consider that traditional observational methods such as RULA or REBA require several dedicated workers for recording, preparation, and data analysis. This incurs a personal time cost, in addition to the salaries associated with these workers, which could be significantly reduced with automated methods as proposed through inertial sensors. Furthermore, two of the main goals for which IMUs have emerged as a method for assessing occupational ergonomics are to mitigate human error from observational methods and reduce economical costs. Therefore, the cost of observational methods is also significant from a personal perspective. However, we have clarified it in the text to avoid confusing the reader (line 373).

5.In lines 359 and 427-428 you would see the issue with using the term “transparency” again. Based on line 359, the problem with IMU can be “lack of transparency”? but in lines 427-428 it has been mentioned that to “improve transparency” (which can infer as to increase transparency) fewer sensors should be used, which may confuse the readers.

Answer: We understand that although we have previously described the term “transparency” in the paper, its use may be confusing when discussing "lack of transparency" or "improving/worsening transparency." Therefore, we have decided to be consistent, correcting throughout the text the instances of such nature, solely using quantitative criteria [i.e., high/low transparency, or increasing/decreasing transparency] (lines 127, 377, 431 and 445).

NOTE: Regarding the English translation, we have translated the work through a specialized service with professional native English-speaking translators. We are attaching the certificate in PDF.

Reviewer 4 Report

Comments and Suggestions for Authors

The article reviews inertial sensors used at work and their impact on ergonomics. It is well prepared, but before publication it is necessary to expand the information related to group citations, e.g. [33-36] or [39-41].

Table 2 provides information on the number of sensors used. Can the authors add information about what sensors are they?

Author Response

Dear reviewer, thank you very much for your comments and suggestions; they have helped us improve the quality of the work. Below, I will address each of the questions you raised one by one:

1.The article reviews inertial sensors used at work and their impact on ergonomics. It is well prepared, but before publication it is necessary to expand the information related to group citations, e.g. [33-36] or [39-41].

Answer: Following your advice, we have expanded the information related to group citations or removed the non-essential references (lines 70-84, 98, 108-121).

2.Table 2 provides information on the number of sensors used. Can the authors add information about what sensors are they?

Answer: Of course, even though all the sensors are inertial units, we have added the available information about that to the table to clarify (Table 2).

NOTE: Regarding the English translation, we have translated the work through a specialized service with professional native English-speaking translators. We are attaching the certificate in PDF.

Round 2

Reviewer 2 Report

Comments and Suggestions for Authors

No further comments.

Reviewer 4 Report

Comments and Suggestions for Authors

After improving the paper is ready to be published. Figure 1 can be placed in the proper place during proofreading.